# Novel Drugs and Radiotherapy in Relapsed Lymphomas: Abscopal Response and Beyond

**DOI:** 10.3390/cancers15102751

**Published:** 2023-05-13

**Authors:** Salvatore Perrone, Paolo Lopedote, Vitaliana De Sanctis, Ilenia Iamundo De Cumis, Alessandro Pulsoni, Paolo Strati

**Affiliations:** 1Department of Hematology, S.M. Goretti Hospital, Polo Universitario Pontino, 04100 Latina, Italy; 2Department of Medicine, St Elizabeth’s Medical Center, Boston University, Boston, MA 02135, USA; 3Department of Radiation Oncology, Faculty of Medicina e Psicologia, Sant’Andrea Hospital, University of Rome “La Sapienza”, 00185 Rome, Italy; 4Department of Radiation Oncology, A. Businco Hospital, ARNAS G. Brotzu, 09121 Cagliari, Italy; 5Department of Lymphoma and Myeloma, The University of Texas MD Anderson Cancer Center, Houston, TX 77030, USA

**Keywords:** abscopal effect, Hodgkin lymphoma, non-Hodgkin lymphoma, radiotherapy, car-t, checkpoint inhibitor, brentuximab vedotin

## Abstract

**Simple Summary:**

Relapsed or refractory (r/r) lymphomas represent hard-to-treat cancers with dismal prognoses. Over the past few years, immune checkpoint inhibitors targeting programmed cell death protein 1 (PD-1) and programmed cell death ligand 1 (PD-L1) have changed treatment paradigms in many malignancies, and are currently approved for Hodgkin lymphoma. Despite improvements in treatment outcomes and the implementation of combined modality approaches, outcomes in patients with r/r lymphomas remain suboptimal. Accumulating evidence suggests that under certain conditions, radiation may be delivered, in conjunction with checkpoint inhibitors or other small molecules, to augment efficacy. In this review, we discuss the interactions of novel therapies approved for lymphomas, mechanisms of synergy with radiation, and how changing the dose, fractionation, and field of radiation may alter the outcomes of these patients.

**Abstract:**

Combined modality has represented a mainstay of treatment across many lymphoma histologies, given their sensitivity to both multi-agent chemotherapy and intermediate-dose radiotherapy. More recently, several new agents, including immunotherapies, have reshaped the therapeutic panorama of some lymphomas. In parallel, radiotherapy techniques have witnessed substantial improvement, accompanied by a growing understanding that radiation itself comes with an immune-mediated effect. Six decades after a metastatic lesion regression outside the irradiated field was first described, there is increasing evidence that a combination of radiotherapy and immunotherapy could boost an abscopal effect. This review focuses on the mechanisms underlying this interaction in the setting of lymphomas, and on the results of pivotal prospective studies. Furthermore, the available evidence on the concomitant use of radiotherapy and small molecules (i.e., lenalidomide, venetoclax, and ibrutinib), as well as brentuximab vedotin, and chimeric antigen receptor (CAR) T-cell therapy, is summarized. Currently, combining radiotherapy with new agents in patients who are affected by lymphomas appears feasible, particularly as a bridge to anti-CD19 autologous CAR T-cell infusion. However, more studies are required to assess these combinations, and preliminary data suggest only a synergistic rather than a curative effect.

## 1. Introduction: Radiation Therapy and Mechanisms of Immunomodulation

Radiotherapy (RT) is an important component of lymphoma treatment for both its curative purpose as well as for palliation [1]. In addition to the direct damage to irradiated cells through the direct breakage of DNA and generation of reactive oxygen species (ROS), ionizing radiation (IR) induces a cascade of events with significant immune-modulatory implications. After exposure to IR, the dying neoplastic cells release damage-associated molecular patterns (DAMPs), including but not limited to calreticulin, high-mobility-group protein B1 (HMGB1), and double-stranded DNA (dsDNA) [2]. More specifically, the translocation of calreticulin to the cell surface of dying cancer cells promotes phagocytosis by dendritic cells (APCs), hence facilitating the subsequent presentation of neoplastic antigens [3]. Conversely, HMGB1 acts as a death signal, and results in the activation of bystander monocytes via the Toll-like receptor 9 pathway as well as endothelial cells, promoting the secretions of more inflammatory cytokines and the recruitment of more immune cells [4,5]. Furthermore dsDNA, released into the cytosol within exosomes from the irradiated cells, is promptly internalized by APC and stimulates cyclic GMP-AMP synthase (cGAS), which in turn leads to the activation of stimulator interferon genes (STING) [6]. The cGAS-STING signaling is a master regulator for the production of type I IFNs within APC, which promotes the maturation and activation of APC and the expansion and survival of CD8+ T cells [7]. Besides releasing type I IFN, cGAS-STING also leads to the activation of the NF-κB signaling pathway, which stimulates APCs by the enhanced expression of IL-12 and CCR7, enhancing their tumor-associated antigens (TAAs) presented to lymphocytes, and priming immune system activity toward the antitumor response [8,9,10] (Figure 1). This change in the cytokine milieu and immune infiltration of the irradiated tissue is responsible for a reshaping effect of IR in the tumor-associated microenvironment (TAM), potentially shifting an immunologically “cold” disease to become a hot disease [11].

In summary, ionizing radiation (IR) can induce the antitumoral immune response through multiple mechanisms, including upregulation of molecular surface proteins such as major histocompatibility complex class I (MHC-1), secretion of pro-inflammatory cytokines, release of neoplastic antigens, and generation of neoantigens. All of these events result in increased antigen presentation by APCs and subsequent recruitment and activation of CD8+ T-cells, which ultimately leads to stronger immune-mediated antitumoral action. The capacity of RT to generate and/or potentiate an antitumoral immune response is corroborated by the possibility to elicit a response in distant, out-of-field localizations in patients undergoing RT, a process defined as the abscopal effect (Figure 2) [7], as opposed to the bystander effect, which is the effect of IR on cells that are few millimeters away from the irradiated field [12]. The key mechanism of the abscopal effect is the above-described “in situ” vaccination induced by IR, mediated by the release of neoantigens and DAMPs [10]. Thus, a localized IR can elicit a systemic response that is mediated by cytokines and chemokines (TNF a, IL-1, IL-6, MCP-1, IL-8) and activate T cells directed against tumor-specific antigens, infiltrating both irradiated and non-irradiated tumor localizations. However, irradiated tumors have also the potential to counterbalance this cascade through several mechanisms, including the upregulation of PD-1 expression and recruitment of myeloid-derived suppressor cells, dampening the immunologic potential of IR [11].

A promising option for patients with relapsed or refractory (r/r) lymphomas is represented by a combination of RT and immunotherapies (e.g., checkpoint inhibitor and/or chimeric antigen receptor T cell), with the goal of enhancing the immune-modulatory action of IR, counter-acting the immune-suppressive tumoral response and promoting an abscopal response. Indeed, while RT allows for immune system activation against epitopes derived from radio-induced cell death, immunotherapy can contrast the immunosuppressive effect of TAM [13,14].

Additionally, given its unique mechanism of action and toxicity profile (e.g., pneumonitis, skin rash, esophagitis), RT has the potential of being combined with other novel agents, resulting in a multi-modal antitumoral action. In the following sections, we summarize the available experiences and ongoing trials (Table 1), exploring RT in combination with innovative treatment options in patients with r/r lymphomas.

## 2. Radiotherapy and Checkpoint Inhibitors: Boosting the Abscopal Effect

A key step to the complex antitumoral action of our immune system is the recognition by T-cell effectors of cancer cells through the interaction of T-cell receptors (TCRs) with MHC-1 + neoplastic antigen. This process is regulated by the ratio of co-stimulatory/co-inhibitory signals [17]. The most important co-inhibitor molecules on T lymphocytes are cytotoxic T-lymphocyte antigen 4 (CTLA4), programmed cell death protein 1 (PD1), B- and T-lymphocyte attenuator (BTLA), lymphocyte activation gene 3 (LAG3), CD160, and the PD1 homolog (PD1H). Their activation by specific membrane ligands leads to the recruitment of intracytoplasmic phosphatases to reverse activation-induced phosphorylation events [18]. Notwithstanding, the tumoral tissue has the ability to impair immune responses, creating an immunosuppressive microenvironment [19,20]. Therefore, with the broad term “checkpoint inhibitors”, we refer to those monoclonal antibodies that antagonize co-inhibitor receptors for their specific ligands, and have the potential to unleash an immune response against cancer cells.

The programmed death (PD) 1 receptor is the most extensively studied receptor involved in the peripheral T-cell inhibitor co-stimulator complex [21]. It is expressed on activated B and T cells, macrophages, DCs, and monocytes. It has two distinct ligands, PD-L1 (CD274, B7H1) and PD-L2, that are expressed on APC and in several cancer subtypes, and in chronic viral-infected cells [22,23,24,25]. Currently, only Hodgkin’s lymphoma (HL) and primary mediastinal B-cell lymphoma (PMBL) [26] have received EMA and/or FDA approval for two anti-PD1 monoclonal antibodies: nivolumab (Nivo) and pembrolizumab (Pembro). Both are humanized, high-affinity IgG4 MOABs that are directed against PD-1 [27,28]. The approval of these agents in the treatment of r/r HL relied on the results of the phase 2 trial Checkmate-205, and the phase 3 study KEYNOTE-204, which showed an overall response rate (ORR) of 63% and superior activity of pembrolizumab over brentuximab vedotin, respectively [29,30]. However, only pembrolizumab is currently approved for r/r PMBL (only in the United States), following an ORR of 45–48% observed in phase 1b of KEYNOTE-013 and in phase II of KEYNOTE-170 trials [31]. The activity of anti-PD1 agents has instead been very limited in DLBCL [32]; among 121 patients with r/r DLBCL ineligible for auto-HCT (*n* = 34) or who relapsed/progressed after auto-HCT (*n* = 87) treated with nivolumab, 10 patients achieved a response (8.2%), complete in 3, with a median PFS of 1.9 months [33]. Similar results have been observed with the use of pembrolizumab as a consolidation strategy after auto-HCT in 29 DLBCL patients, with a PFS rate of 59% at 18 months, lower than the protocol-specified primary objective [34]. The biological mechanisms subtending the limited activity of anti-PD1 agents in DLBCL [35] may reside in the composition of the tumor microenvironment of large B-cell lymphoma (LBCL), encompassing the “effacement” pattern for DLBCL, where the malignant cells proliferate rapidly without dependence on the microenvironment. Conversely, in HL, the infrequent malignant Hodgkin Reed–Sternberg cells attract an extensive supportive milieu of non-malignant cells, generating a “recruiting” pattern [36]. In fact, HL patients frequently carry 9p24.1 amplification, with associated PD-1 ligand overexpression and concurrent gains in the PDL1 and PDL2 loci, supporting the use of an anti-PD1 blockade in this disease [37].

The potential for PD1 inhibitors to boost the rare abscopal effect after radiotherapy has inspired several ongoing trials that are investigating the concurrent administration of PD1-inhibitors and radiotherapy (Table 1). Of interest, a French case series of four patients with heavily pre-treated cHL showed the feasibility of combined PD1 inhibitors (with nivolumab and pembrolizumab in two patients each) with mediastinal radiotherapy, at a median dose of 31 Gy (range: 22–36 Gy), fractioned in about 27 days. While ll patients developed grade 1–2 lung toxicity (resolved after antibiotic therapy or corticosteroids), after 13 months, all were alive and with a complete metabolic response [38]. In another series, two patients with r/r HL were treated with nivolumab and radiotherapy, and achieved long-term CR [39]. Finally, in a case series including 12 patients with r/r HL treated with nivolumab (*n* = 9) or pembrolizumab (*n* = 3) and concomitant local radiotherapy (median RT dose was 30 Gy, range 30–40 Gy], 2 Gy fractioning), after a median follow-up of 1.5 years, 11 (92%) were alive and in CR, while 1 patient progressed and died of infectious complications [40].

For the German Hodgkin Study Group (GHSG) phase II AERN trial, which is investigating the occurrence of the abscopal effect during nivolumab therapy in r/r HL, the interim results were recently presented [15]. Patients with r/r HL refractory to anti-PD1 therapy and with at least two measurable lesions were included in the study; one lesion was included in the radiotherapy field, and one had to be at least 5 cm away from the prescribed 95%. RT was administered at a dose of 20 Gy in 2 Gy fractions, and nivolumab was administered every 2 weeks at a dose of 240 mg for 1.5 years [41]. At the most recent follow-up, nine patients were evaluable for a response; at the first interim restaging after six infusions, five (56%) patients had an abscopal response, and five (56%) achieved a PR (ORR 56%), although the ORR among patients experiencing an abscopal response was not specified. The 1-year PFS rate was 42.3%, and the 1-year OS rate was 100% [15].

The results of two phase II studies investigating the safety and efficacy of checkpoint inhibitors and radiotherapy in patients with r/r HL and DLBCL were recently presented [42,43]. In 17 patients with NHL (15 with LBCL, 5 of whom had double hit, one with PMCL, and one with NK T-cell lymphoma), with a median number of three previous lines of systemic therapy (range 2–13), were treated with 200 mg of pembrolizumab every 21 days with concomitant radiotherapy, at a median dose of 45 Gy (range 22.5–50). RT was delivered to 1–2 sites of the refractory disease (the median number of prior therapies was three; three patients received prior ASCT and nine received CAR T-cell therapy), with at least one measurable lesion excluded from the RT field to allow evaluation of the abscopal effect. Sixteen patients were evaluable for a response; the ORR was 28%, and the CR rate was 23%. Local control was achieved in 11 (69%) patients, with minimal abscopal response. After a median follow-up of 34 months, the median PFS and OS were 2.7 months and 6.3 months, respectively, and the 2-year PFS and OS rates were 11.8% and 23.5%, respectively [42]. This trial, which was terminated early due to slow accrual, is one of the first prospective examples of the limited abscopal effect in LBCL for the combination of pembrolizumab and RT, emphasizing the scarce immunogenicity and poor response to checkpoint inhibitors in this entity. Further studies investigating different RT regimens combined with novel agents are warranted, however, before completely dismissing this strategy in LBCL.

Seven patients with r/r HL who achieved less than CR to nivolumab monotherapy were included in a phase II study combining very low-dose RT (VLDRT), at the dose of 4 Gy over 2 fractions, and nivolumab administered at 3 mg/kg every 2 weeks or 480 mg every 4 weeks (NCT03495713). The post-VLDRT response evaluation in non-radiated lesions revealed a partial response (PR) in two (33%) patients, SD in two (33%) patients, and PD in two (33%) patients. No subject showed disease progression within the VLDRT fields [43]. These preliminary encouraging results emphasize the importance of exploiting two peculiar sensitivity patterns of HL to both radiotherapy and anti-PD1. Therefore, a possible additive and synergistic mechanism of action is not surprising in this setting.

In conclusion, the role of concomitant radiotherapy and anti-PD1 is promising in r/r HL, with preliminary evidence of a potential abscopal effect in half of the patients. However, the number of patients treated to date is limited, doses and fractioning of radiotherapy are heterogeneous, such as different schemes of combinations with anti-PD1 [13], and further confirmatory studies are needed. Most importantly, it remains an open question of whether all lesions should be irradiated or some lesions should be spared, depending on the abscopal effect, possibly using low-dose radiotherapy [44,45].

## 3. Radiotherapy and Small-Molecule Inhibitors

The number of available oral agents and the approved indications for these molecules in NHL is constantly growing. Among those, the most frequently employed categories are Bruton’s tyrosine kinase (BTK) inhibitors, BCL-2 inhibitors, and immunomodulators. Phosphoinositide 3-kinase (PI3K) inhibitors are now falling out of favor, as a result of new safety analyses launched by the European Medical Agency (EMA) and the Federal Drug Agency (FDA), and the only one still approved is copanlisib in r/r follicular lymphoma (FL). The synergy between these agents and RT relies on the activation of different pathways leading to cell apoptosis. The case of venetoclax (Ven) is exemplary. By causing DNA damage, RT results in the p53-mediated activation of Noxa and PUMA and, in turn, in the inhibition of the anti-apoptotic molecules complementary to BCL-2 (BCL-XL and MCL-1), which are implicated in the resistance to Ven [46,47,48]. In a pre-clinical study, RT in combination with Ven was evaluated in a murine model of germinal center (GC) DLBCL, activated B-cell type (ABC) DLBCL, and MCL [49]. A combination with 6 Gy RT resulted in an increase in survival of 18% compared to either Ven or RT alone in the GC-DLBCL model, and by 58% in the ABC DLBCL model. In the MCL model, 8 Gy RT + Ven was associated with nearly three times longer survival times compared to RT alone. The increased synergy noted in the ABC DLBCL model may partially reside in the different mechanism [50] and in the higher frequency [51] of BCL-2 overexpression compared to its GCB counterpart, as well as in the lower levels of BCL-XL in the ABC model [49]. Despite these intriguing findings, data on RT + Ven in NHL patients are limited. A retrospective study analyzed 80 patients with NHL, (DLBCL (*n* = 32), MCL (*n* = 16), chronic lymphocytic leukemia (*n* = 9), marginal zone and FL (*n* = 6), and primary central nervous system lymphoma (PCNSL) (*n* = 3)), treated with either concurrent or sequential therapy with ibrutinib (Ibr) (*n* = 45), lenalidomide (Lena) (*n* = 23), or Ven (*n* = 6), and a median radiation dose of 30 Gy divided on a median of 14 fractions. The adverse events (AE) were compared to the pivotal studies on these agents, and the concurrent treatment was compared to sequential treatment. A higher rate of grade ≥3 neutropenia was noted with the concurrent administration of Ibr when compared to sequential therapy. For the remaining subgroups, no significant difference was noted in the modes of administration compared to the historical controls. Notwithstanding its obvious limitations, this analysis sheds some light on the safety of novel agents in combination with RT. However, given the heterogeneous population, the response rate was not explored [52]. In another retrospective analysis, salvage with lower-dose RT (4 Gy) over 1 or 2 fractions was assessed in 19 patients with r/r MCL for a total of 98 irradiated sites. Fifty-eight percent of the patients were refractory to Ibr. Of the 98 sites, 74 received concomitant therapy (either Lena + rituximab or Ven + obinotuzumab, or either of the latter alone), which was started prior to RT. Despite 76% of the sites receiving concurrent treatment being refractory to the oral agents, CR was achieved in 79/98 (81%) sites without any RT toxicity [53]. Other experiences involving MCL are anecdotal. In a recent case report (51), a 76-year-old man with an MCL, previously treated with Ibr, and with relapse in the pharynx and eyelid, received salvage with RT 18 Gy in the two involved sites along with Lena, achieving a complete response at the 3-month follow-up [54]. In a case of cutaneous MCL treated with a combination of Ibr and 28 Gy RT divided into 14 fractions, the combination was well tolerated, but resulted in only a transient response of about 2 months [55]. More recently, Ibr and low dose (2 × 2 Gy) RT have been assessed in association with intratumoral injection of the TLR9 agonist SD101 [16]. In this promising phase I/II trial (NCT02927964), among the 20 r/r lymphoma (mostly FL) patients, a 50% ORR was noted (1 CR), and lesions reduction in non-injected/non-irradiated lesions were noted in all patients. The side effects were manageable (mostly infusion reaction), supporting the feasibility of this novel approach. Ibr + RT has also been evaluated in the setting of PCNSL. While RT has historically been an essential part of PCNSL treatment, Ibr has only been recently employed in light of the high frequency of the L265P mutation of MYD88 in this lymphoma subtype [56]. As a result of this mutation, PCNSL relies on B-cell receptor (BCR) signaling for survival, and Ibr has the potential to suppress it by inhibiting the BTK, which lays downstream to the BCR pathway [57]. Although several trials have investigated the potential of Ibr in r/r PCNSL, a combination of Ibr + RT has been assessed in only two studies. In a case series, three patients (two with PCNSL and one with secondary CNSL) received Ibr + RT as salvage, all achieving CR. In two patients (one PCNSL and one secondary CNSL), the responses were preserved at 670 and 427 days, respectively [58]. In another case series, six patients (three PCNSL and three secondary CNSL) received this combination, with 5/6 achieving a CR (of which two were PCNSL) [59]. The pooled results of these two studies in a recent meta-analysis showed an ORR and CRR of 85%, which were higher than either therapy alone, but the numbers were too small to determine the real significance of these findings [60]. The combination of novel agents and RT deserves further study to be validated, in terms of their safety and efficacy potential.

## 4. Radiotherapy and Brentuximab Vedotin

The antibody drug conjugate (ADC) brentuximab vedotin (BV) is composed of the coupling of the anti-microtubule cytotoxic agent monomethyl auristatin E and an anti-CD30 antibody via a protease cleavable linker [61]. This agent is currently approved by the FDA as a monotherapy for the treatment of HL after ≥2 lines of therapy, or following an autologous stem cell transplant, and for previously untreated stage III/IV HL in combination with chemotherapy. Besides being expressed on the hallmark cells of HL, the Reed–Sternberg cells, CD30 is also expressed by a subset of T cells and T-cell lymphomas [62,63,64,65]. As such, BV revealed safety and effectiveness, and was subsequently approved in the treatment of anaplastic large cell lymphoma (ALCL), systemic or cutaneous, CD30+ peripheral T cell lymphoma (PTCL), and CD30+ mycosis fungoides (MF).

Different studies have evaluated the combination of BV-based regimens and RT in the setting of HL [66,67,68]. However, except for an initial report [69], RT has been investigated as an end-of-treatment consolidation rather than administered concomitantly with BV, and more attention revolved around the possibility of sparing patients from RT instead of exploring a synergy between the two. In such regard, in a cohort of high-risk pediatric HL patients treated with a BV-containing regimen, the safety and efficacy of RT limited to sites not achieving CR was investigated [67]. This strategy resulted in a 3-year EFS of 97.4% and an OS of 98.7%. Conversely, Kumar [68] assigned adult early-stage high-risk HL patients with a PET-4-negative response following BV, in association with chemotherapy to either RT consolidation (30 Gy ISRT, 20 Gy ISRT, or 30 Gy consolidation-volume RT) or no RT. No significant difference was noted among the different cohorts, with a 2-year PFS of 97% for those not receiving RT (versus 94% for the entire study population). With the growing role of checkpoint inhibitors in HL and the goal of minimizing long term toxicity, this association does not seem to be an area of active interest, and it is unlikely to be further explored. On the other hand, one report recently documented a transient CR in a patient with ALCL using BV + RT [70]. Another retrospective analysis instead evaluated the safety of BV + RT in 12 patients with MF, with or without large cell transformation [71]. Ten patients with DLBCL treated with a combination of polatuzumab vedotin and RT were also included. The toxicities were reported for the whole cohort of patients (DLBCL + MF), and the rates of adverse events before and during concomitant RT were compared. BV with concomitant RT was associated with more grade ≥2 hematological toxicity (79% versus 45%), but with similar rates of non-hematological toxicity (ie, fatigue, diarrhea, neuropathy), although the exact rate of the latter was not reported. Notably, the administration of RT did not result in a higher rate or severity of AE, according to the authors. A prospective phase 2 study (NCT05357794) is currently evaluating the concurrent administration of BV and ultra-low-dose total-skin electron beam therapy in patients with MF, and will possibly shed more light on the efficacy of this combination.

## 5. Radiotherapy and CAR-T

Autologous anti-CD19 chimeric antigen receptor (CAR) T-cell therapy is currently approved by the FDA for the treatment of patients with relapsed or refractory large B-cell lymphoma (LBCL) in second line and beyond [72,73,74,75,76]. While its use has significantly improved outcomes for these patients, the cure rates are limited to 40%, independently from the employed product and/or line of therapy, prompting the need to improve its efficacy. Due to its favorable effects on T cells and myeloid cells, radiation therapy has the potential to achieve such a goal, with possible applications as a bridging, conditioning, and/or consolidation therapy [77,78,79].

Bridging therapy is defined as the treatment provided between leukapheresis and the initiation of conditioning chemotherapy; the role of IR in bridging therapy has been reviewed elsewhere [14,80]. While some of the pivotal trials did not allow it, selecting for a more favorable biology, in real-world practice it is required by the majority of patients, although a standard regimen has not yet been identified [81,82]. The achievement of a low tumor burden and potential complete response with the use of bridging therapy is crucial, as the latter has been shown to be associated with better outcomes after CAR T-cell therapy [83,84,85,86]. However, mere dimensional decrease is not sufficient, as residual disease can still be enriched with pro-tumoral myeloid cells that are responsible for the suppression of CAR T-cell amplification and function [87,88]. Low doses of radiation have been shown to favor the polarization of tumor-associated macrophages to a more antitumoral phenotype, suggesting the possibility of using radiation as a bridging strategy [79]. Two small cases series, including 12 and 5 patients, showed that radiation, at a median dose of 20–30 Gy, can be safely administered as bridging therapy in LBCL patients before CAR T-cells infusion, with no significant increase in cytokine release syndrome, neurotoxicity, and/or cytopenia, as compared to historical data [89,90]. In a larger single-center retrospective study including 124 patients with relapsed or refractory LBCL treated with standard-of-care axicabtagene ciloleucel, 17 patients received bridging radiation, either alone or in combination with systemic therapy. When compared to patients who received systemic bridging therapy alone, those treated with radiation experienced a significantly longer progression-free survival [91]. Studies aimed at identifying the optimal dose, fractionation, and timing of the bridging radiation, and its potential combination with other biological agents, are needed.

Conditioning chemotherapy, which typically includes either fludarabine and cyclophosphamide or bendamustine, is needed for the optimal efficacy of CAR T-cell therapy [92]. While its mechanism of action has historically been associated with an increase in interleukin-7 and 15, secondary to depletion of endogenous lymphocytes, recent data show that its efficacy is also mediated by the eradication of suppressive myeloid cells, including tumor-associated macrophages and myeloid-derived suppressive cells [93,94,95,96,97]. Of interest, the same biological effects of conditioning chemotherapy, including a decrease in regulatory T cells and pro-tumoral myeloid cells, can be achieved with the use of low-dose and ultra-low dose radiation, supporting its use as a conditioning regimen [77,78,79]. In addition, pre-clinical data show that low-dose radiation could increase the sensitivity to CAR T-cell-mediated killing via TRAIL-mediated apoptotic death [78]. In light of the recent international shortage of fludarabine, and of the risk of persistent severe cytopenia associated with the use of conditioning chemotherapy, clinical trials aimed at investigating the safety and efficacy of radiation in this setting are highly needed [98,99,100,101].

Currently, about one third of LBCL patients treated with standard-of-care CAR T-cell therapy achieve only a partial response (PR) to 30 days after cell infusion [72,73,74,75,76]. Although 70% of these will subsequently relapse, the current standard approach is observation, and no consolidation strategy is currently recommended [81,82]. Multiple efforts have been carried out to identify patients in PR who are unlikely to subsequently convert to a complete response (CR), including circulating tumor DNA and PET-based parameters; however, the biology of their disease remains elusive [102,103,104]. Radiation is typically used at a dose of 20–30 Gy to consolidate CR to chemoimmunotherapy in patients with early-stage or bulky advanced-stage LBCL [105,106]. While residual disease is a surrogate marker of refractoriness, and therefore radiation is not typically utilized with curative intent in patients in PR after chemoimmunotherapy, the biology of PR after CAR T-cell therapy may be different, and may benefit from this consolidation strategy. To this regard, case reports have shown that radiation therapy may increase CAR T-cell amplification and function up to months after the initial cell infusion, while also favoring T-cell receptor repertoire expansion [107,108]. Based on these data, consolidative radiation may be considered for patients who only achieve PR after CAR T-cell therapy, though the benefit may be limited to those in whom the residual disease maintains CD19 expression, and to those who received 4-1BB constructs, as the latter are associated with longer persistence [109,110]. Consistent sampling of residual disease among patients with a 30-day response is necessary to better characterize their biology, and to identify optimal targets for the clinical development of consolidative radiation in this setting.

## 6. Discussion

In contrast with patients who achieve durable remissions after first-line therapies, those who experience relapses or, worse, early refractory disease, maintain a dismal prognosis, necessitating the development of new therapeutic approaches. Patients with r/r LBCL benefit from CAR-T cell therapies, when available. Unfortunately, not all patients with LBCL are eligible for CAR-T; for them, alternative options are expanding [111,112]. Similarly, patients with r/r HL benefit from the availability of checkpoint inhibitors (pembro and nivo) and the anti-CD30 drug conjugate brentuximab vedotin. Nevertheless, many patients who are treated with these drugs will not experience a CR, and most will eventually develop disease progression.

In summary, the combination of RT and novel agents appears intriguing in multiple types of lymphoma. This association is of particular interest for histologies that are responsive to checkpoint inhibitors, or are eligible for CAR-T cell. For the first, the AERN trial recently documented a high rate of abscopal response in HL patients previously refractory to anti-PD1 therapy [41]. For patients receiving CAR-T cell, a retrospective study [91] showed that bridging this therapy with RT may result in longer PFS, while studies with a few cases [107,108] have suggested that a consolidation with RT could enhance CAR-T therapy’s amplification and effect. Combinations with other agents remain anecdotal, and more studies are required to clarify their safety and efficacy.

Moreover, multiple questions remain open. The first is about the optimal dose-fractionation regimen to be adopted in patients affected by lymphoma.

Preclinical studies have conveyed conflicting results on whether a high-dose single-fraction approach is superior to a moderate- or low-dose, multiple-fraction approach; some studies have supported the efficacy of a single dose ranging from very low doses of 0.5 Gy [79], 2 to 6 Gy [113], to very high doses of 20–25 Gy [114]. Conversely, other studies have demonstrated that fractionated doses are important for obtaining an abscopal effect (8 Gy × 3 days, or 6 Gy × 5 days) over a single dose of RT [115], and that reaching doses of 10–12 Gy or above may be associated with immunosuppression and loss of the abscopal effect [116,117]. In another study by Potiron et al., an immune-boosting effect was noted with low-dose RT (2 Gy) for a total of 10 days [118]. All of these studies have the limitation of being heterogenous, in terms of their study populations, and focused on patients with a diagnosis other than lymphoma. Lymphomas are radiosensitive tumors that generally respond to fractionations and standard doses [119,120]. In lymphoma patients, most of the cases reporting an abscopal effect described the use of standard RT fraction (2 Gy for each fraction) or a hypofractionated schedule (30 Gy in 10 fr or 20 Gy in 5 fr), especially in the palliative setting [14]. It has also been shown that patients with r/r lymphoma may benefit from accelerated fractionated RT (twice fr per day), especially in case of cross-resistance to chemotherapy and standard RT [121]. Thus, an optimal fractionation has not yet been established.

The second open question remaining is the ideal sequencing of immunotherapy and radiotherapy, and the identification of the ideal agent to combine with RT.

Unfortunately, we have scant data to answer this question; probably, the optimal timing depends on the specific drug used. For example, co-stimulatory agents (e.g., anti-OX40) are best administered rapidly after RT, while anti-PD1 or Anti-CTLA4 could be most effective when administered prior to radiation therapy, in part due to regulatory T-cell depletion, or concomitantly [117,122]. With a high-throughput system enabling the identification of a growing number of agents that could be usefully combined with radiotherapy [123], the problem of the right sequencing remains a major challenge.

Another question remains as to whether as many lesions as possible or only one should be irradiated.

Due to tumor heterogeneity, antigens exposed by irradiation of a single site may not be necessarily shared in non-irradiated sites, or CD8+ T cells may not be able to reach lesions in all sites because of immunosuppressive TAMs. Therefore, a multi-site RT could potentially lead to more antigen presentation and decrease the local immunosuppressive environment, especially in the case of bulky lesions [45]. However, irradiating multiple disease or bulky sites carries the concern of increasing toxicity. Although a definitive answer is not yet available, it is reasonable to modulate our approach based on the patient’s fitness and treatment intent (curative/bridging for more definitive option versus palliative).

The last open question remaining relates to how improvements in radiotherapy techniques and technologies will affect the field.

Potential trials that could help answer these questions may pass through the elucidation of mechanisms of action of radiotherapy in pre-clinical models. For example, humanized mice, transplanted with human umbilical cord blood stem cells and then xeno-transplanted with patient-derived lymphomas, are deemed to be valuable models to study host–lymphoma interactions and immune-modulating agents [124], and radiotherapy could be easily added. Moreover, clinical trials addressing patients who only achieve a partial response after CAR T-cells with a consolidation of radiotherapy, seem feasible and ethically acceptable given the dismal prognosis for these patient populations [125]. Moreover, a combination of the novel and promising bi-specific antibodies (CD3xCD20), e.g., epcoritamab or glofitamab [112,126], with radiotherapy deserves consideration in NHL, but currently no data are available. In HL, the combination of BV and nivo is becoming more investigated [127], and this is another platform that could implement radiotherapy in future trials.

Utilizing modern radiation techniques enables a reduction in treatment volumes and significant sparing of organs at risk, giving the radiotherapist the ability to treat larger diseases (bulky) or more sites of disease. Daily image-guided radiotherapy (IGRT) and intensity-modulated treatments, including static intensity-modulated radiation therapy (IMRT), volumetric-modulated arc therapy (VMAT), and helical therapy (HT), allow conforming of the dose to the target, with significant sparing of healthy organs. When the target is positioned in the chest or abdominal cavity, respiratory-gated delivery strategies, such as deep inspiration breath holding, may minimize dosages to visceral organs such as the lungs, liver, kidney, and heart, hence lowering the incidence of acute events [119,120].

## 7. Conclusions

In conclusion, the association of RT and novel agents used in the modern treatment of patients with lymphomas, albeit intriguing, deserves further study within clinical trials before being routinely adopted. In particular, implementing RT in patients treated with checkpoint inhibitors or with CAR-T cell therapy represents a promising strategy for patients with r/r HL and LBCL, respectively. While several studies, both in the pre-clinical and clinical stage setting, will be required to answer all of the remaining open questions, the way is being paved to a successful combination of modern radiotherapy and biological, cellular, or immunotherapy in patients with lymphomas.

## Figures and Tables

**Figure 1 cancers-15-02751-f001:**
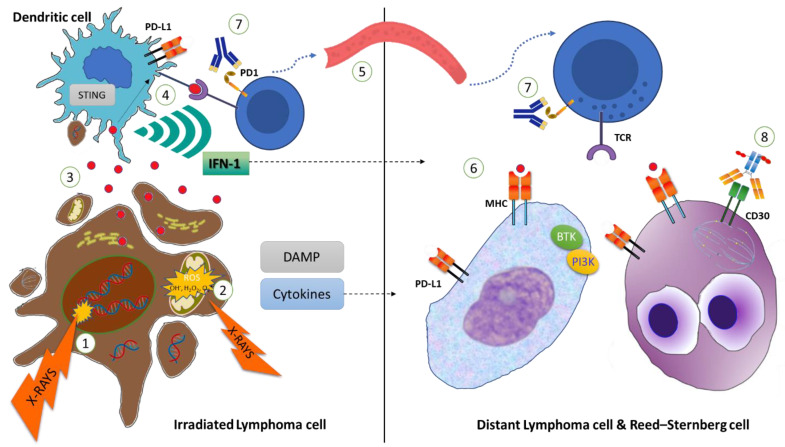
Ionizing radiation on the lymphoma cells leads to nuclear DNA double-strand (ds) breaks (1) and mitochondrial damage (2), which in turn increases intracellular reactive oxygen species (ROS) levels, ultimately resulting in apoptosis of the lymphoma cells (Left). Cancer-associated antigens and damage-associated molecular patterns (DAMPs), including dsDNA, are released from the dying cells, and are internalized by dendritic cells (or APC) (3). dsDNA stimulates the activation of interferon stimulator genes (STING), and cancer-associated antigens are presented to CD8+ T cells (4). In parallel, STING leads to Interferon-1 (IFN-1) and other cytokine secretions, which in turn stimulate the expansion and survival of CD8+ T cells (4) in the irradiated site, as well as in more distant non-irradiated sites (dotted line). Activated CD8+ T cells can migrate through the bloodstream (5) and recognize cancer-associated antigens, displayed within major histocompatibility complex (MHC)-1 (6) by non-Hodgkin lymphoma (NHL) cells or Reed–Sternberg (RS) cells (Right). The cytotoxic effect of CD8+ lymphocytes can be potentiated by systemic treatment, including anti-PD1 monoclonal antibodies (7), BTK-inhibitors, or antibody drug conjugates as brentuximab vedotin (8), targeting CD30 on RS cells and interfering with mitotic fuse.

**Figure 2 cancers-15-02751-f002:**
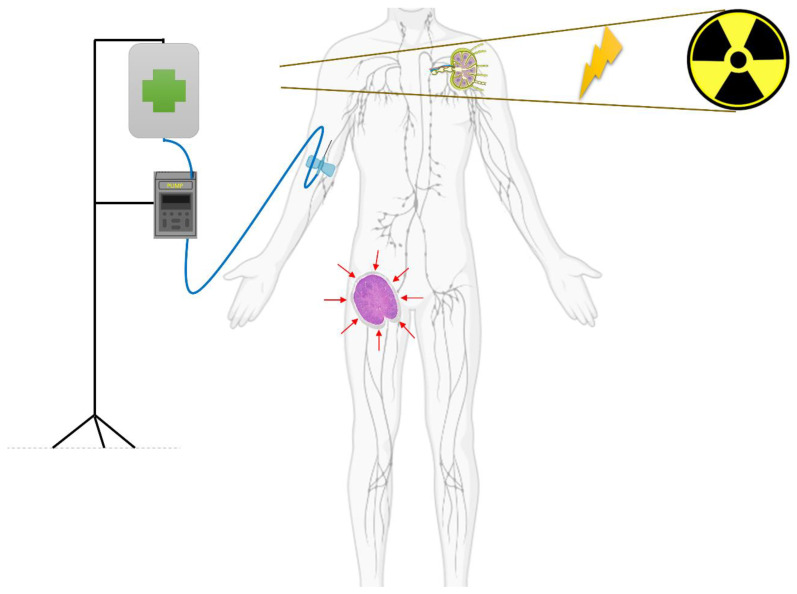
Abscopal effect. A patient with a lymphoma is receiving an immuno-modulatory drug (infusion pump is depicted), and is concurrently receiving radiation therapy to his left axillary lymph node. Outside the field of radiotherapy, his right inguinal lymph nodes experience a response (red arrows suggesting reducing dimensions of the lymph node), thanks to the immune system’s activity (T-cell activation).

**Table 1 cancers-15-02751-t001:** Clinical trials currently evaluating radiotherapy in combination with checkpoint inhibitors.

NCT Number	Disease (Previous Lines)	Treatment	Phase	Status
NCT04417166	NK/T Cell Lymphoma (0)	Pembrolizumab + RT	II	Recruiting
NCT04827862	NHL (≥2)	Pembrolizumab + Low-Dose RT	II	Recruiting
NCT03480334	HL (≥1 anti-PD1 agent)	Nivolumab + RT	II	Recruiting [15]
NCT03179917	HL (≥1)	Pembrolizumab + ISRT	II	Recruiting
NCT04962126	FL (0)	Atezolizumab + Obinotuzumab +/− RT	II	Recruiting
NCT03210662	NHL (≥1)	Pembrolizumab + EBRT	II	Recruiting
NCT03610061	DLBCL or FL (≥1)	Durvalumab + RT	I	Active, not recruiting
NCT02927964	FL, MZL, MCL	SD-101 + Ibrutinib + RT	I/II	Active, not recruiting [16]
NCT04365036	ENKTL, nasal type (0)	Chemo-RT +/− Toripalimab	III	Recruiting
NCT04676789	ENKTL, nasal type (0)	Sintilimab + PegAsp + RT	II	Not yet recruiting
NCT04366128	ENKTL, nasal type (0)	Camrelizumab + PegAsp + Apatinib + RT	NA	Recruiting
NCT05477264	ENKTL, nasal type (0)	Tislelizumab + RT	II	Not yet recruiting
NCT05149170	ENKTL, nasal type (0)	Tislelizumab + RT	II	Recruiting

NK = natural killer. RT = radiation therapy. NHL = non-Hodgkin’s lymphoma. HL = Hodgkin’s lymphoma. FL = follicular lymphoma. ISRT = involved-site radiation therapy. DLBCL = diffuse large B-cell lymphoma. ENKTL = extranodal natural killer/T-cell lymphoma. MZL = marginal zone lymphoma. MCL = mantle cell lymphoma.

## Data Availability

No new data were created or analyzed in this study. Data sharing is not applicable to this article.

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
