# Peer review of "Novel Drugs and Radiotherapy in Relapsed Lymphomas: Abscopal Response and Beyond"

_cancers, 2023, doi:10.3390/cancers15102751_

Round 1

Reviewer 1 Report

The authors report an interesting review on the possibile role of low dose radiotherapy to improve clinical results of check-point inhibitors, targeted therapy and CAR-T in lymphoma patients. Although these new approaches have been limited until now to few patients, they are promising and very interesting. The review looks complete and well written.

Author Response

Thanks to the reviewer for positive comments!

Reviewer 2 Report

I asked the Editor to diregard my review. 

Author Response

Thanks for the comment. It is difficult to add something.

Reviewer 3 Report

Ferrone et al provide a review on the use of newer agents (small molecule inhibitors, Brentuximab, CAR-T and immune checkpoint in combination with radiotherapy, providing a theoretical basis for their combination.

SIMPLE SUMMARY AND ABSTRACT: both accurately summarize the text of the paper.

INTRODUCTION: the introduction provides background with regard to the utility and mechanisms of action of radiation therapy and targeted therapies, checkpoint inhibitors, brentuximab and CAR-T. These are probably provided into four separate sections discussing each for systemic therapy with radiation therapy, providing a factual background and references.

DISCUSSION: The discussion appropriately approaches the combinations of radiation and small molecule inhibitors, checkpoint inhibitors, brentuximab and CAR-T,   mostly addressing the findings in the introduction as requiring further investigation (“open questions”). The authors should discuss briefly, perhaps in one paragraph, the design of potential clinical trials to address these open questions.

CONCLUSION: The conclusion accurately summarizes the discussion, and should also contain a statement about future clinical trial design reflecting that in the discussion.

FIGURES AND TABLES: Appropriate and not redundant.

REFERENCES: All are appropriate for subject matter.

ADDITIONAL: The numbering is confusing. The introduction is labeled as 1, in the separate sections regarding different systemic therapies is 2-5. The discussion is 4 and the conclusion is 5. This requires revision or explanation..Recommend revision to add on recommendations for designs of future clinical trials.

Author Response

We re-numbered the paragraphs.

We entered some clues for future trials in the discussion in a brief paragraph.

A sentence about trials was also added to the conclusion.

Thank for your suggestions, they improoved the manuscript